# Disturbance of the Conformation of DNA Hairpin Containing the 5′-GT-3′ Binding Site Caused by Zn(II)bleomycin-A$_5$ Studied through NMR Spectroscopy

**Kyle L. Covington and Teresa Lehmann \***

Department of Chemistry, University of Wyoming, Laramie, WY 82071, USA

\* Correspondence: tlehmann@uwyo.edu; Tel.: +1-307-766-2772

**Abstract:** The antibiotics known as bleomycins constitute a family of natural products clinically employed for the treatment of a wide spectrum of cancers. These antibiotics have the ability to chelate a metal center, most commonly Fe(II), and cause site-specific DNA cleavage upon oxidation. Bleomycin therapy is a successful course of treatment for some types of cancers. However, the risk of pulmonary fibrosis as an undesirable side effect, limits the use of the antibiotics in cancer chemotherapy. Bleomycins are differentiated by their C-terminal, or tail, regions, which have been shown to closely interact with DNA. Pulmonary toxicity has been correlated to the chemical structure of the bleomycin C-termini through biochemical studies performed in mice. In the present study, we examined the binding of Zn(II)Bleomycin-A$_5$ to a DNA hairpin of sequence 5′-CCAGTATTTTTACTGG-3′, containing the 5′-GT-3′ binding site. The results were compared to those from a previous study that examined the binding of Zn(II)Bleomycin-A$_2$ and Zn(II)Peplomycin to the same DNA hairpin. We provide evidence that, as shown for DNA hairpins containing the 5′-GC-3′ binding site, Zn(II)BLM-A$_5$ causes the most significant structural changes to the oligonucleotide.

**Keywords:** DNA; NMR; pulmonary fibrosis; anticancer drug; structure–function

---

## 1. Introduction

Bleomycins (BLMs) (Figure 1) are a group of glycopeptide antibiotics isolated from *Streptomyces verticillus*, which are widely used for the treatment of numerous neoplastic diseases [1]. Clinically, BLMs are employed for the treatment of squamous cell carcinomas [2], non-Hodgkin's lymphomas [3], testicular carcinomas [4] and ovarian cancer [5]. The drug acts as an antitumor agent by the ability of a metal complex of the antibiotic to cleave DNA [6]. Since the introduction of the clinically used mixture of BLMs, Blenoxane, to clinical medicine in 1972, attempts have been made at modifying the basic BLM structure at the C-terminus to improve its therapeutic index. However, other than their demonstrated role in binding to DNA, the pharmacological and toxicological importance of particular tails on BLM remains unclear. Over 300 different BLM analogs have been screened in attempts to find species that combine high antitumor activity with low pulmonary toxicity [7–10].

When administered in chemotherapy, BLM can lead to the development of pulmonary fibrosis in some patients. The mechanism by which the therapeutic use of BLM is frequently associated with this side effect is poorly understood. However, previous studies on pulmonary toxicity caused by BLM have linked the BLM C-terminus with the severity of BLM-induced lung injury in mice [11–17]. These studies have also identified BLM-A$_2$ and -A$_5$ as very toxic, and BLM-B$_2$ and peplomycin (PEP) less toxic to mice. On the other hand, correlations between lung-cell toxicity and anticancer activity by BLM are more tenuous. For instance, PEP has been found to exhibit stronger anticancer activity, with lower

pulmonary toxicity [7,8]. CuBLM-A$_2$ has been reported as having stronger antitumor activity than CuBLM-B$_2$ against Ehrlich carcinoma and sarcoma 180 [18]. Phleomycin obtained from the culture of filtrate of *Streptomyces verticillus* and separated into various compounds were found to exhibit different degrees of antibacterial, anticancer, and lung-damaging activities [19].

**Figure 1.** Structures of BLM-A$_2$, BLM-A$_5$, and PEP. BLM, bleomycin; PEP, peplomycin.

We have performed a series of studies designed to determine the effect of four different metallo-BLMs (MBLMs) [Zn(II)BLM-A$_2$, -A$_5$, -B$_2$, and PEP] on the structure of DNA hairpins of sequences 5′-AGGCCTTTTGGCCT-3′ (OL$_1$) and 5′-CCAGTATTTTTACTGG-3′ (OL$_2$), containing the BLM-preferred 5′-GC-3′ and 5′-GT-3′binding sites, respectively [20–22]. The results of these studies indicated that, when bound to a 5′-GC-3′ site, Zn(II)BLM-A$_5$ had the most significant effect on the conformation of OL$_1$ [20]. Additionally, the network of connectivities exhibited by free Zn(II)BLM-A$_5$ was also modified to the greatest extent when bound to this hairpin [22]. In the present study, we examine the effect of the binding of Zn(II)BLM-A$_5$ on the conformation of OL$_2$, and compare this effect with those produced by the binding of Zn(II)BLM-A$_2$ and Zn(II)PEP previously published [21]. BLM-A$_5$ is a very toxic BLM, and the fact that it could significantly alter the conformation of hairpins containing 5′-GC-3′ and 5′-GT-3′ binding sites could have a correlation with its toxicity.

We provide evidence that the formation of the Zn(II)BLM-A$_5$-OL$_2$ triad renders the most significant modifications on the conformation of OL$_2$.

## 2. Results

The broadening and shifting of the imino signals in OL$_2$ is an indication of the drugs' binding to OL$_2$. In a one-dimensional NMR spectrum derived from a DNA segment, the imino signals appear in the region between 11 and 15 ppm, which is devoided of other signals. This fact makes these signals ideal to determine if a ligand is bound to the DNA segment. Upon ligand binding, a complex bigger in size than the native DNA segment will form, and the tumbling time of this new complex will be longer than that of the free DNA. It is expected then that the imino signals will broaden and shift from their original positions after complexation. As seen in Figure 2, not all of the BLMs studied produce the same effect on the imino-region protons. Binding of Zn(II)BLM-A$_2$ produces the most significant broadening of the imino signals in OL$_2$, followed by Zn(II)PEP and Zn(II)BLM-A$_5$. As previously reported by us [20–22], these shift differences are due to the different C-terminus on each of the different MBLMs.

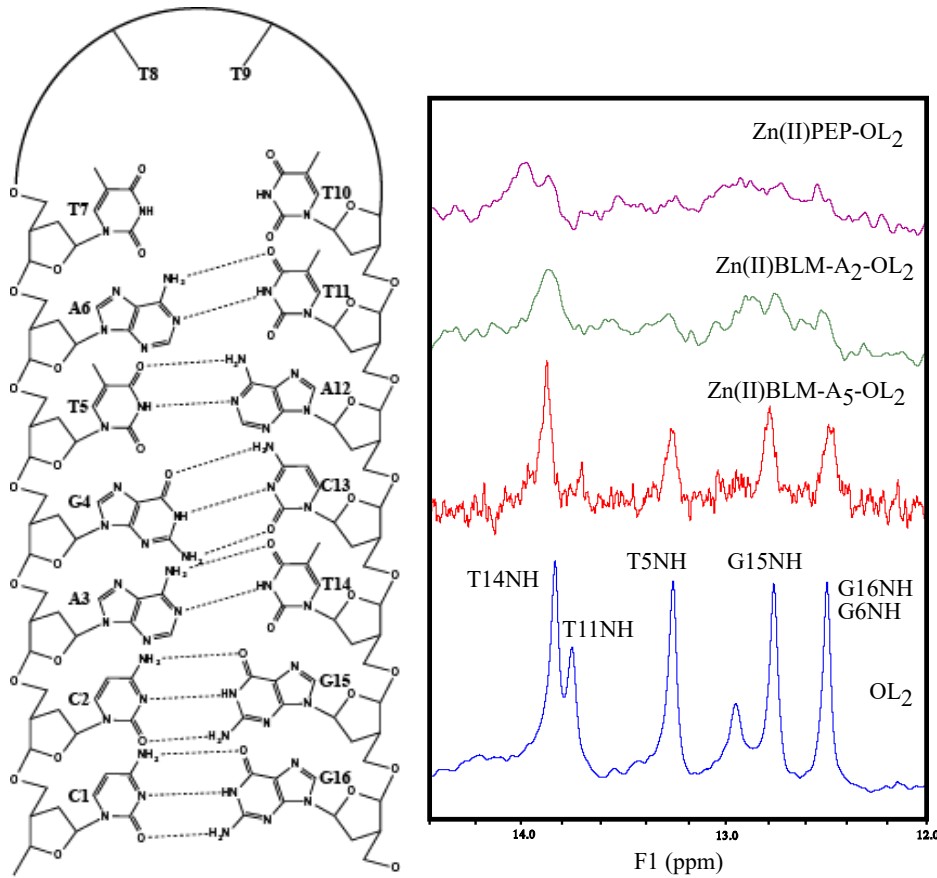

**Figure 2.** (**Left**): Native OL$_2$. (**Right**): Imino region of the 1D $^1$H-NMR spectra in H$_2$O collected at 5 °C for free OL$_2$ and OL$_2$ bound to Zn(II)BLM-A$_2$, -A$_5$, and Zn(II)PEP. Zn(II)BLM:DNA molar ratio 1:1.

NOESY spectra of free and Zn(II)BLM-bound OL$_2$ were collected to compare the chemical shifts of the exchangeable-proton signals of OL$_2$ in both conditions. These spectra were acquired at 5 °C to allow unambiguous assignments of the NH and NH$_2$ protons in OL$_2$. The NH and NH$_2$ signals in OL$_2$ are labeled in Figure 3, and the corresponding assignments are shown in Table S1. Overlays of the NOESY spectra in H$_2$O collected at 5 °C for free OL$_2$ (black) and OL$_2$ bound to Zn(II)BLM-A$_2$, -A$_5$, and Zn(II)PEP (red) are shown in Figure 4; Figure 5. Signal labeling is the same as in Figure 3. Table 1 shows how the imino and amino signals in the bound OL$_2$ differ from those in the free form of the hairpin. For all triads, there is significant shift ($\Delta\delta$ bigger than or equal to |0.04|) of the 2b protons in the C2NH$_2$, A3NH$_2$ and A12NH$_2$ groups for OL$_2$. The NH$_2$ protons in base A6 shifted in all triads and overlapped with other signals in the NOESY spectra, hinting significant, unquantifiable, shifts for these protons. Globally, the bases T11 and T14 show significant imino disturbances in Zn(II)BLM-bound OL$_2$, with base T11 displaying significant, unquantifiable shifts in the A$_5$ triad. This triad additionally displays significant shifts for the 2a protons in bases A3 and A12. Comparison of number of protons affected, and the magnitudes of the significant shifts exhibited by all triads indicates that the binding of Zn(II)BLM-A$_5$ has a more notable effect on the inter-strand conformation of OL$_2$ upon MBLM binding.

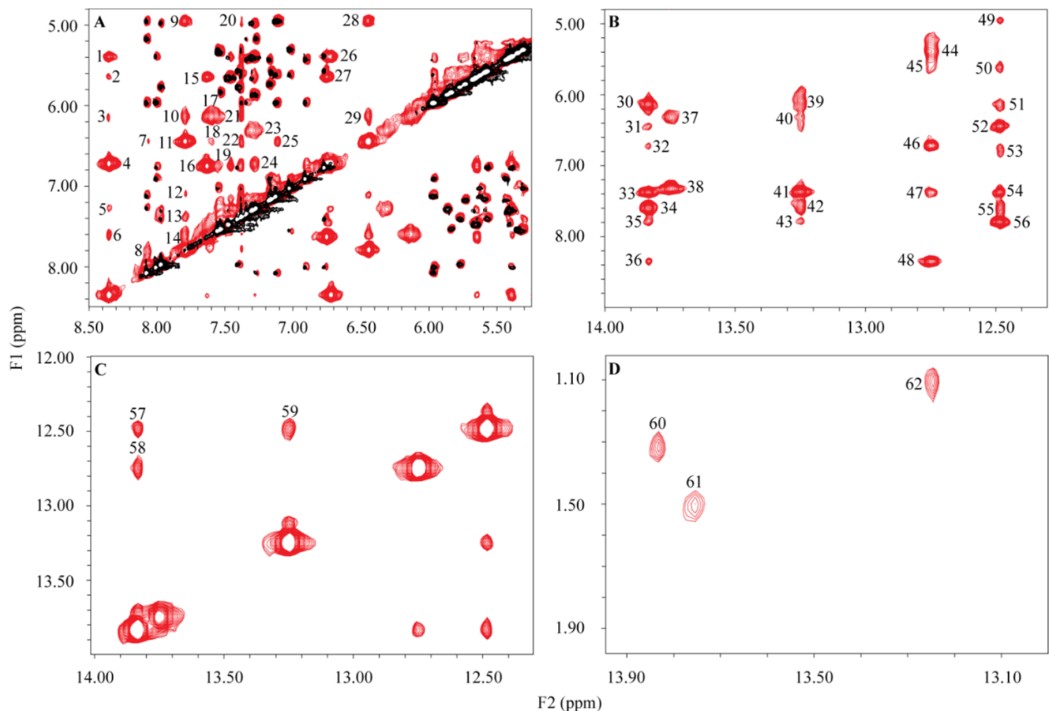

**Figure 3.** OL$_2$ signals in H$_2$O (red) and D$_2$O (black) acquired at 5 °C. (**a**) base and NH$_2$ protons (**b**) and (**c**) imino protons. (**d**) Methyl region. The corresponding assignments are listed in Table S1.

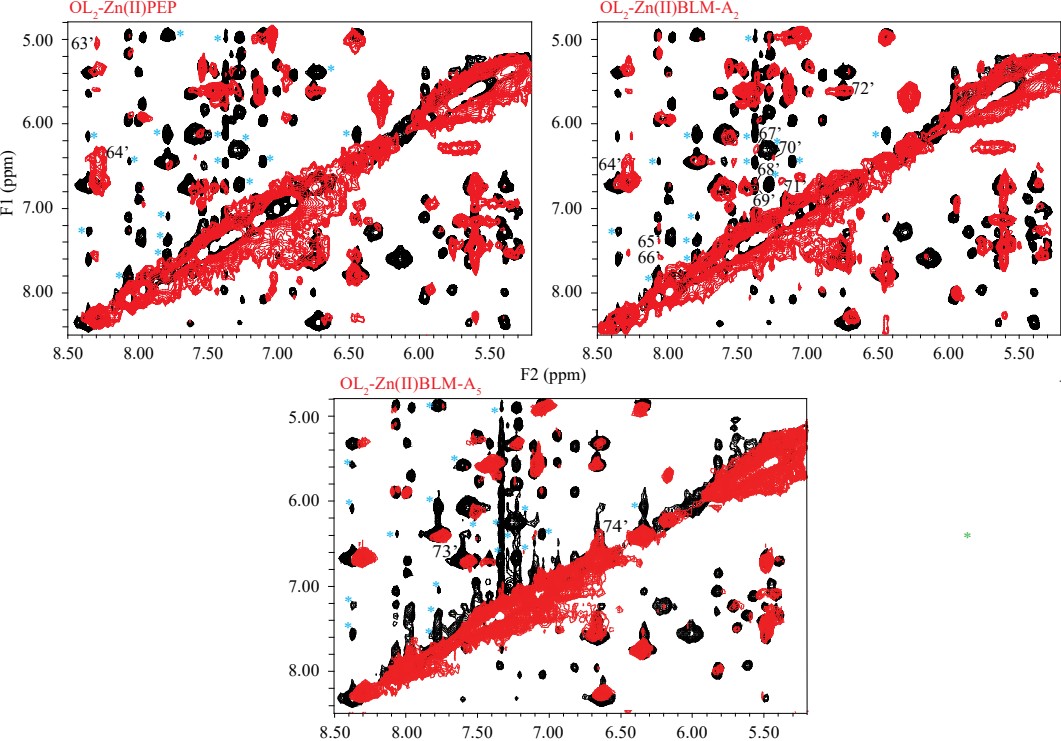

**Figure 4.** Overlays of the amino region of the NOESY spectra for samples in H$_2$O acquired at 5 °C for free OL$_2$ (black) and OL$_2$ bound to each Zn(II)BLM (red). The missing NOE signals are indicated by astericks (*) and are identified in Table S1. New NOE signals are labeled with (') and their corresponding assignments are listed in Table S1.

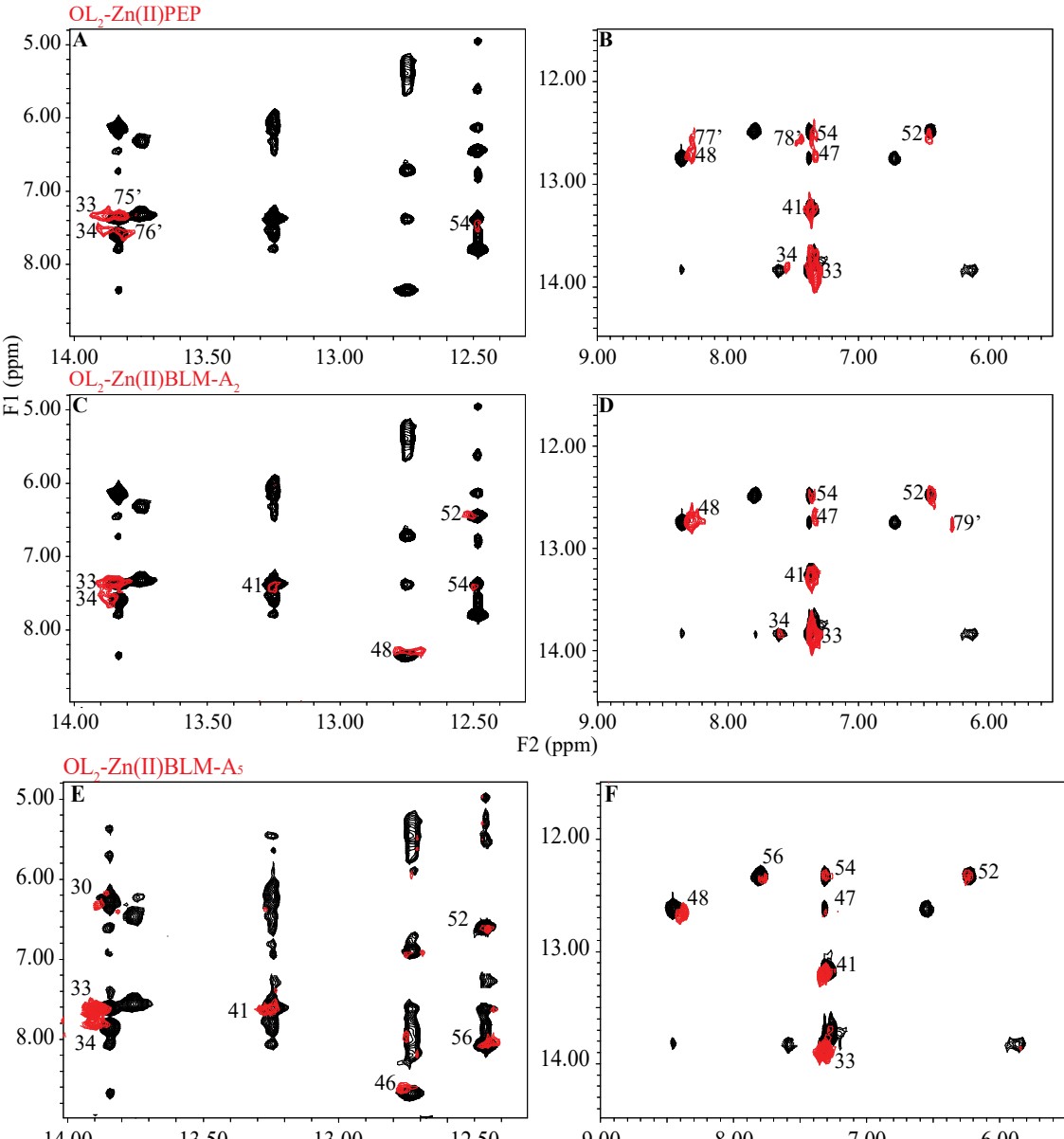

**Figure 5.** Overlays of the imino region of the NOESY spectra for samples in $H_2O$ at 5 °C for free $OL_2$ (black) and $OL_2$ bound to each Zn(II)BLM (red). Free-$OL_2$ NOE signals that were conserved upon MBLM binding are labeled with the same numbers used in Figure 3. New NOE signals are labeled primes. The assignments of all these signals is listed in Table S1.

As can be seen in Figures 4 and 5, binding of Zn(II)BLMs to $OL_2$ leads to the disappearance of some of the NOEs detected for exchangeable protons in free $OL_2$ (indicated by asterisks in Figure 4, and compiled in Table S1), indicating that the distances between the corresponding protons become greater than 5 Å upon MBLM complexation. Some of the missing NOEs are common to all triads, but there are also differences in the missing NOEs depending on the bound MBLM. The data presented in Figures 4 and 5 have been summarized in schematic representations of free $OL_2$ and Zn(II)BLM-bound $OL_2$ (Figure 6) for easier visualization. For free $OL_2$, 62 NOEs were detected in $H_2O$ at 5 °C [20]. Figure 6B–D shows the NOEs displayed by $OL_2$ that are conserved when a particular Zn(II)BLM is bound. This figure also shows the new NOEs that were detected for the bound $OL_2$. Examination of Figure 6 indicates that many of the lost NOEs are on the plane of the base pairs. Many of the shorter interactions between those base pairs where binding occurs are missing. This result suggests that these planes are highly disturbed upon Zn(II)BLM binding, making the backbone of the $OL_2$ more unstable.

Figure 6 shows that the number of conserved NOEs by the $A_5$ triad (19, 31%) is less than that in the PEP (22, 35%) and $A_2$ (24, 39%) triads [21]. These results hint that Zn(II)BLM-$A_5$ binding affects the inter-strand region of $OL_2$ to a greater extent than Zn(II)PEP and Zn(II)BLM-$A_2$. Similar results were gathered from Zn(II)BLM-$A_5$ binding to a DNA hairpin containing the 5′-GC-3′ binding site [20].

**Table 1.** Differences in chemical shift displayed by the $NH_2$ and NH protons in free and metallo-BLMs (MBLM)-bound $OL_2$ for samples in $H_2O$ at 5 °C.

| | Chemical Shift (F2 (ppm)) | | | | | | |
|---|---|---|---|---|---|---|---|
| Assignment | $OL_2$ | $OL_2$-Zn(II)PEP | $\Delta\delta$ [a] (PEP) | $OL_2$-Zn(II)BLM-$A_2$ | $\Delta\delta$ [a] ($A_2$) | $OL_2$-Zn(II)BLM-$A_5$ | $\Delta\delta$ [a] ($A_5$) |
| $C1NH_{2b}$ | 7.63 | 7.61 | 0.02 | 7.64 | −0.01 | 7.58 | 0.05 |
| $C1NH_{2a}$ | 6.75 | 6.75 | 0.00 | 6.77 | −0.02 | 6.75 | 0.00 |
| $C2NH_{2b}$ | 8.36 | 8.29 | 0.07 | 8.28 | 0.08 | 8.20 | 0.16 |
| $C2NH_{2a}$ | 6.72 | 6.66 | 0.06 | 6.70 | 0.02 | 6.72 | 0.00 |
| $A3NH_{2b}$ | 7.60 | 7.56 | 0.04 | 7.54 | 0.06 | 7.76 | −0.16 |
| $A3NH_{2a}$ | 6.13 | 6.11 | 0.02 | 6.15 | −0.03 | [b]_ | [b]_ |
| G4NH | 12.48 | 12.51 | −0.03 | 12.51 | −0.02 | 12.45 | 0.03 |
| T5NH | 13.25 | 13.27 | −0.02 | 13.25 | 0.00 | 13.26 | −0.01 |
| $A6NH_{2b}$ | 7.28 | [b]_ | [b]_ | [b]_ | [b]_ | [b]_ | [b]_ |
| $A6NH_{2a}$ | 6.31 | [b]_ | [b]_ | [b]_ | [b]_ | [b]_ | [b]_ |
| T11NH | 13.75 | 13.82 | −0.07 | 13.86 | −0.11 | [b]_ | [b]_ |
| $A12NH_{2b}$ | 7.60 | 7.56 | 0.04 | 7.54 | 0.06 | 7.58 | 0.04 |
| $A12NH_{2a}$ | 6.13 | 6.11 | 0.02 | 6.15 | −0.01 | 6.73 | 0.60 |
| $C13NH_{2b}$ | 7.79 | 7.78 | 0.01 | 7.77 | 0.02 | 7.75 | 0.04 |
| $C13NH_{2a}$ | 6.44 | 6.46 | −0.02 | 6.44 | 0.00 | 6.45 | −0.01 |
| T14NH | 13.83 | 13.87 | −0.04 | 13.86 | −0.02 | 13.88 | −0.05 |
| G15NH | 12.75 | 12.72 | 0.03 | 12.71 | 0.04 | 12.78 | −0.03 |
| G16NH | 12.48 | 12.56 | −0.08 | 12.51 | −0.03 | 12.47 | 0.01 |

[a] Calculated as (chemical shift in free OL)—(chemical shift in bound OL). [b] (-) Unassignable upon complexation with $OL_2$.

The non-exchangeable protons in the DNA bases conform the core of the $OL_2$ hairpin. The manner in which the non-exchangeable protons shift in the NMR spectra collected upon complexation with different Zn(II)BLMs is also worth examining. Figure 7 shows the base region of NOESY spectra collected at 25 °C in $D_2O$ for free $OL_2$ (black) and $OL_2$ in the presence of Zn(II)BLM-$A_2$, Zn(II)BLM-$A_5$, and Zn(II)PEP (red). It can easily be seen in this figure that the degree and direction (upfield/downfield) of shifting of the base signals in bound $OL_2$ varies upon complexation of different BLMs. Table 2 shows the differences in chemical shifts between the free and Zn(II)BLM-bound $OL_2$ for each of the different MBLM-triads. From examination of Table 2, it can be determined that for all triads, there are significant shifts ($\Delta\delta$ greater than or equal to |0.04| ppm) for the bases A3, G4, T7, T9, and C13. The rest of the bases have moderate to small shifts depending on the MBLM bound to the $OL_2$. Three of these five bases show more significant shifts in the $A_5$ triad.

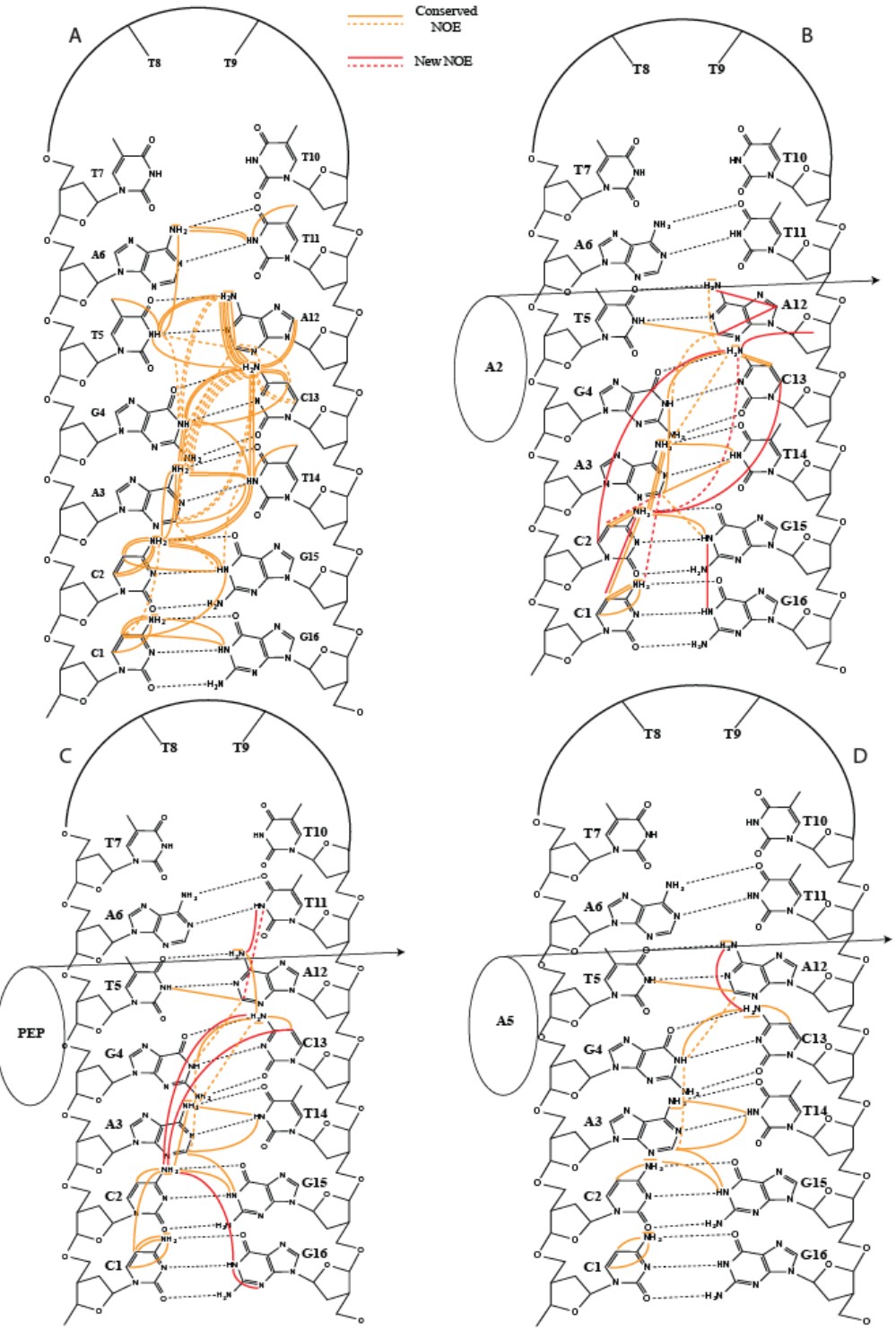

**Figure 6.** Schematic representation of the NOE connections between the exchangeable protons for: (**A**) free $OL_2$. (**B**–**D**) $OL_2$ bound to Zn(II)BLM in $H_2O$ at 5 °C. Dashed and continuous lines both represent NOEs, and are used together to avoid confusion in busy regions of the scheme. The conserved (orange) and new (red) NOE connectivities shown correspond to spectra in Figures 4 and 5 and are also listed in Table S1. The NOE connectivities shown for free $OL_2$ correspond to spectra in Figure 3 and are also listed in Table S1.

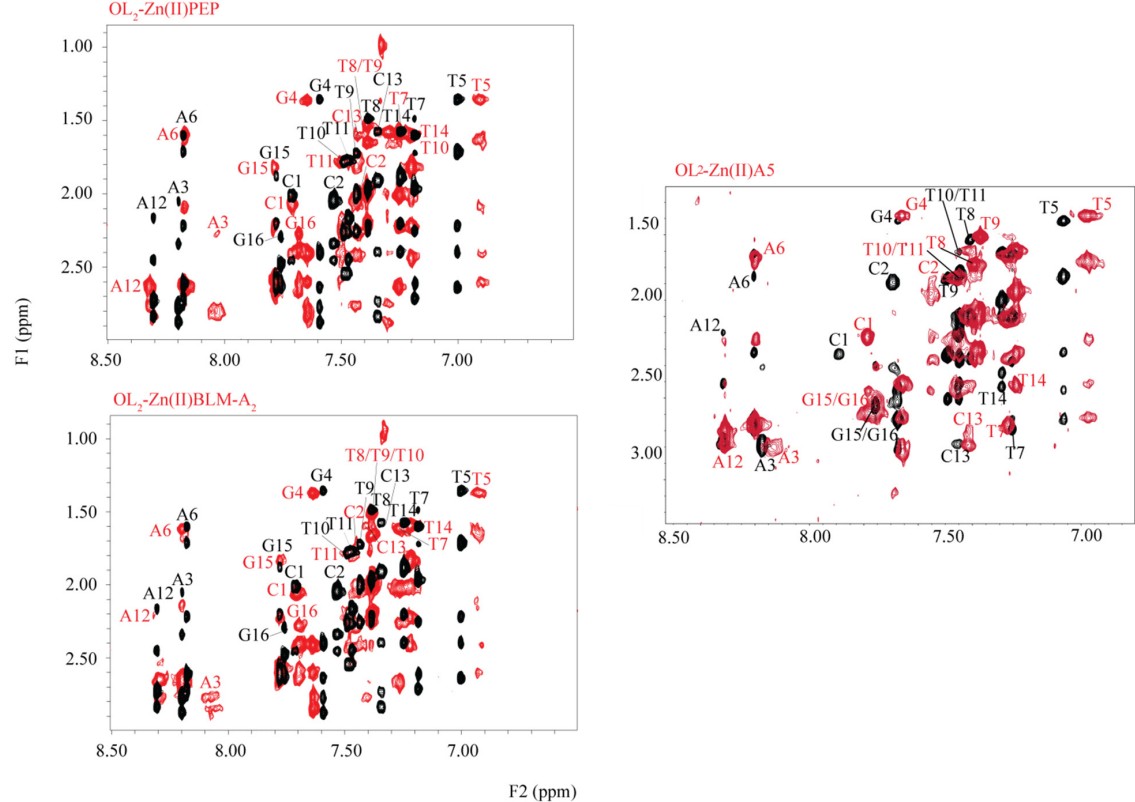

**Figure 7.** Overlays of the base regions of the NOESY spectra for samples in D$_2$O acquired at 25 °C for free OL$_2$ (*black*) and Zn(II)BLM-bound OL$_2$ (*red*).

**Table 2.** Chemical shifts of the base protons in free and Zn(II)BLMs-bound OL$_2$ for samples in D$_2$O at 25 °C.

| | | | Chemical Shift (F2 (ppm)) | | | | |
|---|---|---|---|---|---|---|---|
| Base [1]H | OL$_2$ | OL$_2$-PEP [a] | Δδ [b] (PEP) | OL$_2$-A$_2$ [a] | Δδ [b] (A$_2$) | OL$_2$-A$_5$[a] | Δδ [b] (A$_5$) |
| C1(C6H) | 7.70 | 7.71 | 0.01 | 7.71 | 0.01 | 7.78 | −0.08 |
| C2(C6H) | 7.53 | 7.43 | 0.10 | 7.46 | 0.07 | 7.55 | −0.02 |
| A3(C8H) | 8.20 | 8.03 | 0.17 | 8.08 | 0.12 | 8.13 | 0.07 |
| G4(C8H) | 7.59 | 7.64 | −0.05 | 7.64 | −0.04 | 7.66 | −0.07 |
| T5(C6H) | 7.00 | 6.90 | 0.10 | 6.93 | 0.07 | 6.97 | 0.03 |
| A6(C8H) | 8.18 | 8.17 | 0.01 | 8.20 | −0.02 | 8.20 | −0.02 |
| T7(C6H) | 7.18 | 7.25 | −0.07 | 7.27 | −0.08 | 7.27 | −0.09 |
| T8(C6H) | 7.39 | 7.39 | 0.00 | 7.39 | 0.00 | 7.39 | 0.00 |
| T9(C6H) | 7.44 | 7.39 | 0.05 | 7.39 | 0.05 | 7.37 | 0.07 |
| T10(C6H) | 7.49 | 7.30 | 0.19 | 7.39 | 0.10 | 7.46 | 0.03 |
| T11(C6H) | 7.47 | 7.50 | −0.03 | 7.47 | 0.00 | 7.46 | 0.01 |
| A12(C8H) | 8.30 | 8.32 | −0.02 | 8.31 | 0.01 | 8.31 | −0.01 |
| C13(C6H) | 7.35 | 7.44 | −0.09 | 7.41 | −0.06 | 7.42 | −0.07 |
| T14(C6H) | 7.25 | 7.20 | 0.05 | 7.22 | 0.03 | 7.24 | 0.01 |
| G15(C8H) | 7.78 | 7.79 | −0.01 | 7.78 | 0.00 | _[c] | _[c] |
| G16(C8H) | 7.76 | 7.68 | 0.08 | 7.70 | 0.06 | 7.75 | 0.01 |

[a] Chemical shift of the proton in OL$_2$ when bound to the corresponding Zn(II)BLM. [b] Calculated as (chemical shift in free OL$_2$)—(chemical shift in bound OL$_2$). [c] Missing NOE signal.

The network of NOE connectivities displayed by the non-exchangeable protons in free OL$_2$ is also disturbed upon complexation of different MBLMs. A spectrum displaying these connectivities is provided in Figure S1, and the NOE connectivities are listed in Table S2. A schematic representation of them is shown in Figure 8. Figure 9 shows overlays of sugar region of the NOESY spectra collected for

free (black) and bound OL$_2$ (red) at 25 °C in D$_2$O. Examination of this figure shows that the protons in the sugar units attached to the DNA bases exhibit not only shifts form their positions in the native hairpin, but also a different network of connectivities, including missing native NOEs and new NOEs (marked with asterisks and primes, respectively, in Figure 9). Schematic representation of the NOE connections detected for MBLM-bound OL$_2$ are shown in Figure 10, and listed in Table S2.

　　Examination of Figure 9 indicates that out of 82 NOEs detected for native OL$_2$ [21], the PEP and A$_2$ triads conserve 66% and 62%, respectively. On the other hand, the A$_5$ triad conserves only 39% of these NOEs. This result is a sign that the binding of Zn(II)BLM-A$_5$ to OL$_2$ affects the conformation of the core of the hairpin more significantly than the binding of the other MBLMs included in this study.

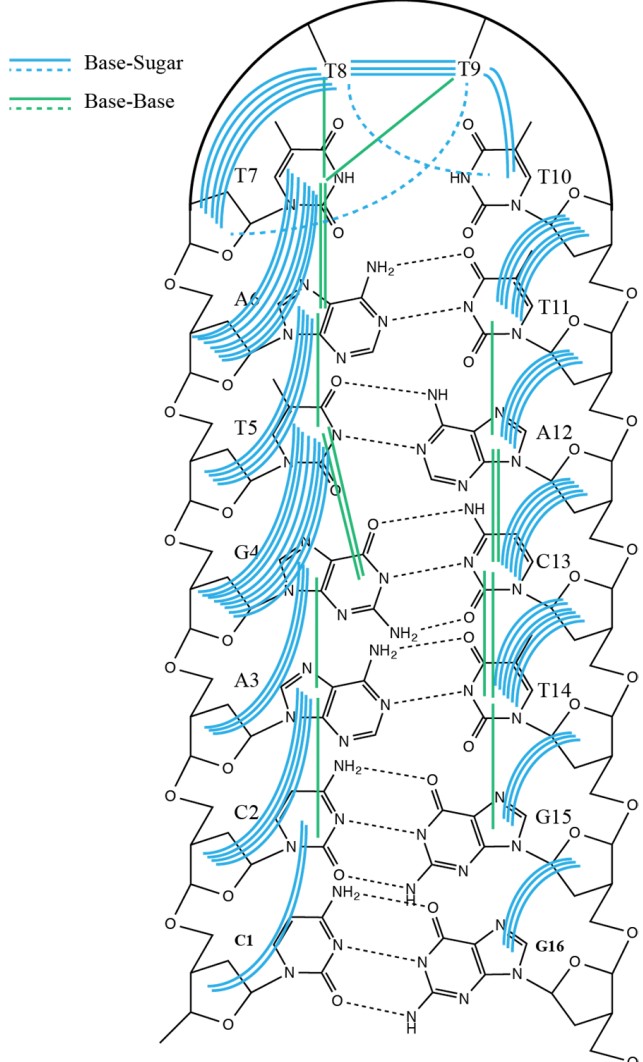

**Figure 8.** Schematic representation of the inter-base NOE connections for free OL$_2$ in D$_2$O at 25 °C. Base–base (green) and base–sugar (blue). NOE connectivities are shown here with lines and listed in Table S2. Dashed lines are used to avoid confusion in the busy sectors of the scheme.

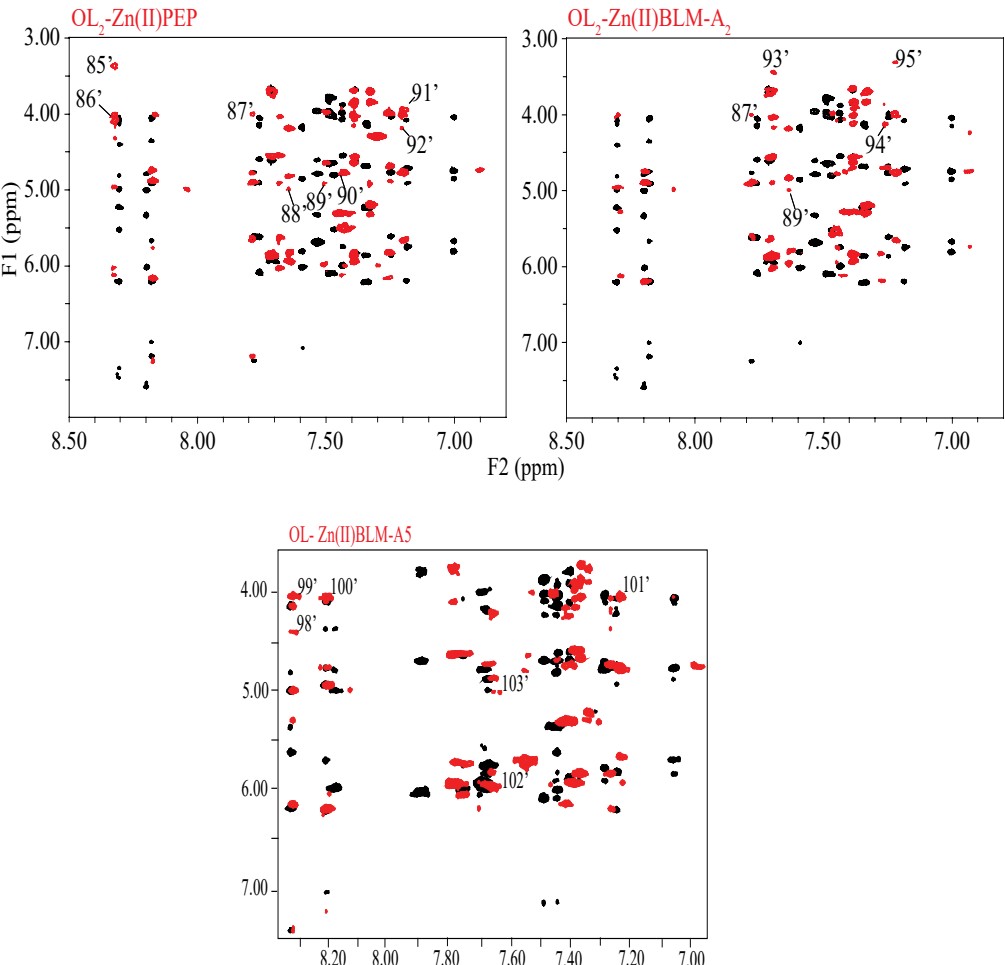

**Figure 9.** Overlays of the NOESY spectra for samples in D$_2$O acquired at 25 °C for free OL$_2$ (black) and Zn(II)BLM-bound OL$_2$ (red). Shown here are the NOEs between the base non-exchangeable protons and their corresponding sugar protons. NOEs only detected in the presence of MBLMs are marked with primes.

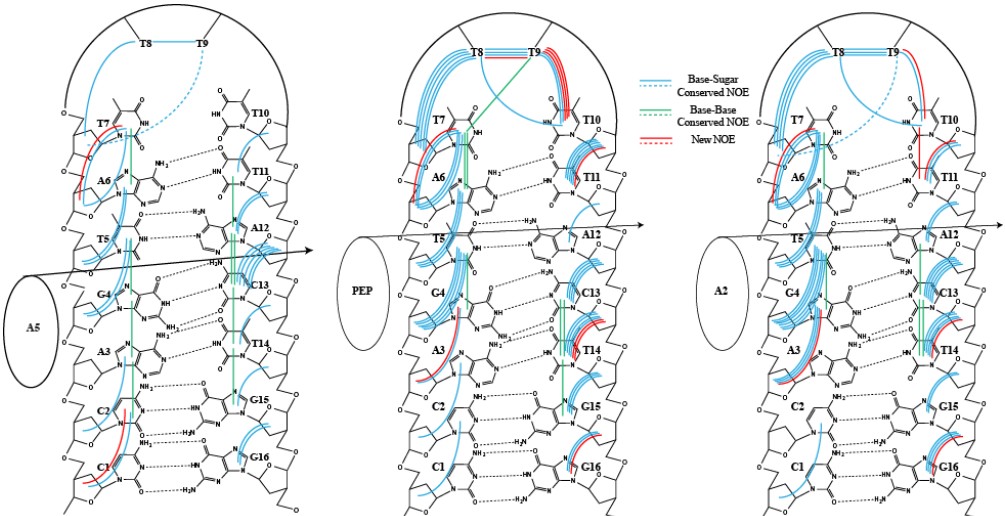

**Figure 10.** Schematic representation of the NOE connections for OL$_2$ bound to MBLM in D$_2$O at 25 °C. Base–base (green), base–sugar (blue), and new (red) NOE connectivities are shown here with lines and listed in Table S2. Dashed lines are used to avoid confusion in the busy sectors of the scheme.

## 3. Discussion

To date, the structures of MBLM-DNA complexes published in the literature [23–34] have been determined using different MBLMs, interacting with different DNA fragments, with a focus on the intercalating capability of the bithiazole-segment and a disregard of the effect of the substituent C-termini upon binding. The absence of a systematic approach to controlling experimental factors such as DNA-base sequences and C-terminus substituents in the aforementioned studies have led to a lack of clarity when it comes to characterizing the solution structure of these drugs, and their authentic interactions with DNA. Methodical studies controlling each of these factors individually are required in order to establish their relative significance.

The study presented herein is the last in a series of research works designed to determine the effect of four different Zn(II)BLMs on the structure of DNA hairpins $OL_1$ and $OL_2$, containing the BLM-preferred 5′-GC-3′ and 5′-GT-3′ binding sites, respectively. The results of these research works indicated that, when bound to a 5′-GC-3′ site, Zn(II)BLM-$A_5$ had the most significant effect on the conformation of $OL_1$ [20]. In the present study, we examined the effect of the binding of Zn(II)BLM-$A_5$ on the conformation of $OL_2$, and compare this effect with those produced by the binding of Zn(II)BLM-$A_2$ and Zn(II)PEP. The data presented in Figures 4 and 5, Table 1 and Table S1 indicate that, among the triads tested, the binding of Zn(II)BLM-$A_5$ has a more notable effect on the inter-strand conformation of $OL_2$ upon MBLM binding. Figure 6 supports this notion, since the $A_5$ triad only conserves 31% of the native NOEs detected for free $OL_2$. Regarding the core structure of $OL_2$, the data displayed in Figures 8–10, Table 2 and Table S2 suggest that the binding of Zn(II)BLM-$A_5$ affects the native core structure of $OL_2$ to the greatest extent among the Zn(II)BLM considered herein. The $A_5$ triad produces some of the highest values for $\Delta\delta$ (Table 2), and only conserves 39% of the native NOEs detected for $OL_2$ (Figures 9 and 10). The loss of NOE signals upon MBLM complexation could also be interpreted on the basis of exchange phenomena taking place between free and Zn(II)BLM-bound $OL_2$. This possibility was explored in the course of the study of Zn(II)BLM-$A_2$ and Zn(II)PEP bound to $OL_2$ [21] (data not shown). The ROESY spectra collected for both triads exhibited no cross peaks due to exchange. Since all BLMs have very similar chemical structure and molecular weight, we assume that ROESY spectra collected for the $A_5$ triad would render the same results. In the absence of exchange ROE signals, we interpret the missing NOE signals as a sign of changes in the conformation of $OL_2$ upon MBLM complexation.

Intermolecular interactions in the form of NOEs were not detected either at 25 or 5 °C. Intermolecular NOEs were detected by Manderville et al. [26] for Zn(II)BLM bound to a DNA fragment containing a 5′-GC-3′ binding site only at 35 °C but not at lower temperatures, and the detected intermolecular NOEs were very week signals. All of our spectra were collected at 25 or 5 °C, which made us able to assign NH and $NH_2$ signals. These signals cannot be assigned at higher temperatures due to fast exchange with $H_2O$.

Based on the overall results of the present investigation, we can conclude that the binding of Zn(II)BLM-$A_5$ causes the most significant distortions to the native structure of $OL_2$. The same result was found through the study of Zn(II)BLM-$A_5$ bound to $OL_1$ [20]. It is important to notice that all Zn(II)BLMs considered herein and in previous research [20–22] differ from each other only on the structures of their C-termini (tails). Therefore, the different degrees of distortion produced to the native structures of $OL_1$ and $OL_2$ can safely be attributed to the different tails contained in BLM-$A_2$, -$B_2$, -$A_5$, and PEP. Zn(II)BLM-$A_5$ has been found to produce the most significant distortion to the inter-strand and core native structures of both $OL_1$ [20,21] and $OL_2$ (present investigation). Interestingly, Raisfeld et al. [11–17] have classified BLM-$A_5$ as the most toxic of BLMs regarding pulmonary cytotoxicity. Although the source of BLM pulmonary toxicity has not been linked to its interaction with DNA, it is tempting to propose a possible connection between the level of disturbance of DNA upon MBLM complexation, and that of pulmonary toxicity resulting from the use of different BLMs in cancer chemotherapy. Detailed structural characterization at the molecular level of BLM binding to DNA and the structure/toxicity connection could unveil more intimate details of the

MBLM–DNA triads. These new details could advance the BLM field vertically by guiding the synthesis and/or isolation of specific congeners with tails producing very low or no pulmonary toxicity. These new BLMs could improve the quality of life of cancer patients under BLM regimens. The development of more intensive chemotherapy regimens has increased the incidence of toxic pulmonary side effects following administration of an increasing number of commonly used cytotoxic drugs [35–38], and pulmonary complications are no longer restricted to BLM. It is possible that the structure/toxicity correlations that could be delineated as a result of detailed studies of MBLM–DNA interactions will help to better understand this side effect when caused by other drugs.

## 4. Materials and Methods

BLM-$A_2$ was purchased form TOKU-E (Bellingham, WA, USA). BLM-$A_5$ was purchased form LKT Laboratories, Inc. (St. Paul, MN, USA). PEP was a generous gift of Nippon Kayaku Co., Ltd. (Tokya, Japan). Zinc Sulfate 7-hydrate was purchased from VWR (Radnor, PA, USA). Deuterated water (99.9%, d), sodium hydroxide, and sodium chloride were purchased form Sigma-Aldrich (St. Louis, MO, USA). The oligonucleotide $OL_2$ was purchased from Integrated DNA Technologies, Inc. (Coralville, IA, USA).

### 4.1. NMR Sample Preparation

BLMs, 1.95 μmol, was dissolved in 650 μL of $D_2O$. A 0.12 M solution of $ZnSO_4 \cdot 7H_2O$ was then added to the BLM solutions to afford Zn(II):BLM mole ratios of 1:1. The pH (meter reading uncorrected for the deuterium isotope effect) was adjusted to 6.5 with a 0.1 M NaOD solution. DNA, 0.325 μmol, was dissolved in 585 μL of $D_2O$. Subsequently, 65 μL of a 200 mM NaCl solution was added, and the pH was adjusted to 6.5. Various aliquots of Zn(II)BLM were then added to the DNA sample and the Zn(II)BLM:DNA 1:1 complex formation was followed by monitoring the changes in one-dimensional (1D) $^1$H-NMR spectra. When the DNA to Zn(II)BLM ratio becomes 1:1, no additional changes are observed in the 1D spectra of the samples. All Zn(II)BLM–DNA (triad) samples were prepared at Zn(II)BLM:DNA molar ratios 1:1. Zn(II)BLM samples in 90% $H_2O$/10% $D_2O$ (referred to as spectra in $H_2O$) were prepared by analogous procedures.

### 4.2. NMR Spectra Collection

NMR experiments were performed at 600 MHz on a Bruker AVANCE III 600 spectrometer (Bruker BioSpin Corp, Billerica, MA, USA) with a 5.0 mm multi-nuclear broad-band observe probe. Spectra were collected at 278 MK (samples in $H_2O$), and 298 K (samples in $D_2O$), and referenced to HDO and $H_2O$ as the internal standards. For the two-dimensional total correlation spectroscopy (TOCSY), and two-dimensional nuclear Overhauser effect spectroscopy (NOESY) experiments, solvent suppression was achieved using excitation sculpting with gradients. The mixing times in NOESY and TOCSY experiments were typically 200 and 40 ms, respectively. Spectral widths were typically 10 ppm (samples in $D_2O$), and 18 ppm (samples in $H_2O$) in both dimensions, and 512 $t_1$ points were acquired with 2048 complex points for each free induction decay (FID). The number of scans for a $t_1$ point was usually 32. Spectra were Fourier transformed using Lorentzian-to-Gaussian weighting and phase-shifted sine-bell window functions. Processing and analysis of the NMR data were performed using Topspin3.0 and NMRViewJ software.

### 4.3. NMR-Signal Assignments

The assignment of the NMR signals coming from protons in the oligonucleotide was performed following the protocols developed by Kurt Wüthrich [39]. The NMR signals coming from Zn(II)BLM were assigned based on the methodology followed by Akkerman et al. [40].

**Supplementary Materials:** The following are available online at http://www.mdpi.com/2312-7481/5/3/52/s1, Figure S1: NOESY spectrum of free OL$_2$ in D$_2$O collected at 25 °C, Table S1: NOESY connectivities displayed by OL$_2$ in H$_2$O at 5 °C, Table S2: Inter-base NOE connections detected in D$_2$O at 25 °C for free and MBLM-bound OL$_2$.

**Author Contributions:** K.L.C. prepared the NMR samples, collected NMR data, analyzed and interpreted NMR spectra and participated on the writing of this manuscript, T.L. provided the research idea, supervised and managed the project, and participated in data interpretation and manuscript writing.

**Funding:** This research was funded by National Institute of Health [Grant 1R15GM106285-01A1].

**Acknowledgments:** This work was supported in whole by the National Institute of Health [Grant 1R15GM106285-01A1]. Our gratitude also goes to Nippon Kayaku Co., Ltd. (Tokyo, Japan) for the generous gift of peplomycin. We also acknowledge Alexander Goroncy and Sally Murray for help collecting the NMR data presented in this work.

**Conflicts of Interest:** The authors declare no conflict of interest.

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
