# Peer review of "Disturbance of the Conformation of DNA Hairpin Containing the 5′-GT-3′ Binding Site Caused by Zn(II)bleomycin-A5 Studied through NMR Spectroscopy"

_magnetochemistry, doi:10.3390/magnetochemistry5030052_

Round 1

Reviewer 1 Report

In this work, the authors analyze by NMR spectroscopy the effect of the Zn(II)beleomycin-A5 with a DNA hairpin. It is an interesting research, in the effort of anticancer drug development. However, in my opinion this work should be improved. Above all, a more detailed and better discussed NMR characterization and assignments, also with homo or heteronuclear correlation experiments of the studied complexes could be useful. Moreover, could the authors better describe Figure 2?  

Figure 1 was already reported in the literature, please cite it in the caption.

I found several typos in the text (i.e,line 7, "sates", please correct. line 21, after "hairpin", add "."; line 58; "to the "grates"??;  line 92 "acquried", change with "acquired") please revise.

Once improved I would suggest this paper for publication in Magnetochemistry.

Author Response

Reviewer 1

Thank you very much for your comments.

I will cite this Reviewer’s comments textually in italics and bold, and follow with an answer to each of them.

Comment 1:“    in my opinion this work should be improved. Above all, a more detailed and better discussed NMR characterization and assignments, also with homo or heteronuclear correlation experiments of the studied complexes could be useful.”

Answer 1: The assignment of the NMR signals coming from the oligonucleotide was performed following the protocols developed by Kurt Wüthrich in his book “NMR of Proteins and Nucleic Acids”. These protocols are the standard way oligonucleotide signals are identified, and are very extensive to detail in any manuscript. The assignment of the signals coming from Zn(II)BLM were performed using the methodology followed by Akkerman et al. Detailing this methodology would add a lot to the text of the manuscript, and would not contribute to its main goal. An extra section was added to the manuscript in the Material and Methods section to indicate these facts (lines 287-291).

Comment 2: “Moreover, could the authors better describe Figure 2?

Answer 2: In a one-dimensional NMR spectrum derived from a DNA segment, the imino signals appear in the region between 11 and 15 ppm, which is devoided of other signals. This fact makes these signals ideal to determine if a ligand is bound to the DNA segment. Upon binding, a complex bigger in size than the native DNA segment will form, and the tumbling time of this new complex will be longer than that of the free DNA. It is expected then that the imino signals will broaden and shift from their original positions after complexation.

This paragraph has been added to the manuscript to better explain the meaning of figure 2 (lines 66-71).

Comment 3: “Figure 1 was already reported in the literature, please cite it in the caption.

Answer 3: Reference numbers were added to the caption of figure 1 to indicate that it has been reported in the literature (line 38).

Comment 4: “I found several typos in the text (i.e,line 7, "sates", please correct. line 21, after "hairpin", add "."; line 58; "to the "grates"??; line 92 "acquried", change with "acquired") please revise.

Answer 4: All corrected. Thank you for noticing.

Reviewer 2 Report

    The manuscript by K.L. Covington and T. Lehmann reports the NMR  results of binding of Zn(II)Bleomycin-A5 to DNA hairpin containing 5’-GT-3’ binding site, OL1. The authors compare the chemical shift changes upon  Zn(II)Bleomycin-A5 complexation to OL1 to the earlier study that examined the binding of Zn(II)Bleomycin-A2  and Zn(II)Peplomycin to the same oligonucleotide (J. Biol. Inorg. Chem. (2017) 22:1039–1054). Additionally, analyzing the previously published results on Zn(II)Bleomycin-A5, Zn(II)Bleomycin-A2  and Zn(II)Peplomycin binding to DNA hairpin with 5’-GC-3’ binding site, OL2 (J Biol Inorg Chem (2017) 22:121–136), concludethat the binding of Zn(II)Bleomycin-A5 causes the most significant distortions to the native structure of OL2”.

This is already the fourth publication from the series on the same subject. In this regard, this is surprising that authors do not even intend to relate their results to other studies on Bleomycin-DNA interactions, several of which are included as citations. There is no information on how the obtained data correlate with the modes of Bleomycin binding to DNA as proposed by the other author. Vast majority of the results presented in the manuscript has already been published previously, including tables and figures. In summary, the work does not bring sufficient scientific novelty and I do not recommend this work for publication in Magnetochemistry.

Author Response

Reviewer 2

Thank you very much for your comments.

I will cite this Reviewer’s comments textually in italics and bold, and follow with an answer to each of them.

Comment 1: “This is already the fourth publication from the series on the same subject. In this regard, this is surprising that authors do not even intend to relate their results to other studies on Bleomycin-DNA interactions, several of which are included as citations. There is no information on how the obtained data correlate with the modes of Bleomycin binding to DNA as proposed by the other author.

Answer 1: One of the facts we hope we have shown through the studies depicted in the present manuscript, and the other papers in the series, is that the NMR results derived from a particular triad (metal-BLM-DNA), depend on the type of BLM (A2, A5, B2, and PEP) and the type of DNA segment used (GC or GT binding sites). That is, different DNA segments bound to different BLMs produce different NMR results. These facts make comparison with the work of other authors pointless, unless the authors have used the same BLM and DNA segment that we have used. This is the main reason why comparisons are not included in our manuscripts. In fact, we strongly believe that another factor that should be tested is the influence that the metal center bound to BLM has on the structure of triads.

Comment 2: “Vast majority of the results presented in the manuscript has already been published previously, including tables and figures. In summary, the work does not bring sufficient scientific novelty and I do not recommend this work for publication in Magnetochemistry.

Answer 2: We disagree with your opinion, with all due respect. NMR data on Zn(II)BLM-A5 bound to a hairpin containing the 5’-GT-3’ binding site has not been presented before. Granted that the data on Zn(II)BLM-A2 and Zn(II)PEP was already published (ref 21), as indicated in the manuscript, but we need to use that data to compare to what happens when Zn(II)BLM-A5 binds the same hairpin. The binding of Zn(II)BLM-A5 to a hairpin containing the 5’-GC-3’ binding site studied in ref 20, indicated that this metallo-BLM is the one that alters the hairpin’s structure the most. Our goal in the present manuscript is to determine if the same results are found when a 5’-GT-3’ binding site is tested. BLM-A5 is a very toxic BLM, and the fact that it could significantly alter the conformation of hairpins containing GC and GT binding sites could have a correlation with its toxicity.

We have added the last two lines of the paragraph above to the introduction of the manuscript to make our goal more evident (line 59-60).

Reviewer 3 Report

This is an interesting manuscript exploring the molecular details of Metallobleomicins to DNA hairpins.

I have one major issue that I would like the authors to consider and a couple of minor issues.

General concern, the authors do not show evidence of having explored the possibility that the mobility, including exchange phenomena, is different in the presence of MBLM and this would offer an alternative interpretation for the missing nOes in addition to the structural distortion discussed in the manuscript. I suggest that this alternative is also discussed.

Minor issues

Labels in figure 2 are too small

Line 81 how did the authors reach the conclusion that 0.04 delta shift is the minimum for significant changes

Figure 3 Asterisks need to be of a different color to make them more visible. I suggest blue that is easy for most color blind people

Figure 6 shows intrabase nOes (for example T5 or T11) which are not expected to be disturbed upon binding by MBLM. I suggest their removal.

Figure 7 shows conserved and new nOes. Can one infer that the remaining differences from figure 6 represent missing nOes? Given that an important part of the discussion is based on new and missing nOes  I suggest reporting the missing nOes explicitly either by adding a new color to figure 7 or by making figure 7 with just the new and missing nOes and placing the conserved nOes in supplementary materials.

Typos

Line 58, probably “greatest”

Line 204 probably “data”

Author Response

Reviewer 3

Thank you very much for your comments.

I will cite this Reviewer’s comments textually in italics and bold, and follow with an answer to each of them.

Comment 1: “General concern, the authors do not show evidence of having explored the possibility that the mobility, including exchange phenomena, is different in the presence of MBLM and this would offer an alternative interpretation for the missing nOes in addition to the structural distortion discussed in the manuscript. I suggest that this alternative is also discussed.”

Answer 1: Exchange effects were not explored for the binding of Zn(II)BLM-A5 to OL2 in the present manuscript. However, in our work with Zn(II)PEP and Zn(II)BLM-A2 bound to the same oligonucleotide (ref 21) ROESY spectra were collected to explore this possibility. The ROESY spectra collected for both triads exhibited no cross peaks due to exchange. Since all BLMs have very similar chemical structure and molecular weight, we assume that ROESY spectra collected for the A5 triad would render the same results. In the absence of exchange ROE signals, we interpret the missing NOE signals as a sign of changes in the conformation of OL2 upon MBLM complexation. A paragraph stating these results has been added in the reviewed manuscript as part of the discussion (lines 216-223).

Comment 2: “Labels in figure 2 are too small

Answer 2: The size of the labels has been increased.

Comment 3: “Line 81 how did the authors reach the conclusion that 0.04 delta shift is the minimum for significant changes

Answer 3: As a standard in NMR spectroscopy, Dds between 0.04 and 0.05 ppm are considered sufficient to indicate significant changes in the chemical/magnetic environment of a proton.

Comment 4: “Figure 3 Asterisks need to be of a different color to make them more visible. I suggest blue that is easy for most color blind people

Answer 4: There are no asterisks in Figure 3. I assume you mean Figure 4. All of the asterisks in Figure 4 were changed to blue.

Comment 5: “Figure 6 shows intrabase nOes (for example T5 or T11) which are not expected to be disturbed upon binding by MBLM. I suggest their removal.”

Answer 5: We respectfully disagree. In our opinion, all native NOEs need to be considered when studying the binding of a ligand to a target. Changes in sites different from the binding site can occur, and they would not be accounted for if some NOEs were not considered.

Comment 6: “Figure 7 shows conserved and new nOes. Can one infer that the remaining differences from figure 6 represent missing nOes? Given that an important part of the discussion is based on new and missing nOes I suggest reporting the missing nOes explicitly either by adding a new color to figure 7 or by making figure 7 with just the new and missing nOes and placing the conserved nOes in supplementary materials.

Answer 6: The answer to the first question in the paragraph above is yes. In our opinion, presenting the conserved and new NOEs shows better the contrast between the free and bound oligonucleotide. Adding or only including the missing NOEs to Figure 6 would make it very busy, and we would have to do the same in Figure 10. Additionally, we have adopted this style of data presentation in this series of four papers. Changing the style, would make linking the papers very difficult for the readers. The missing NOEs are compiled in the supplementary materials.

 “Line 58, probably “greatest”

Corrected. Thank you.

Line 204 probably “data”

Corrected. Thank you.

Reviewer 4 Report

In the manuscript by Covington and Lehmann the interactions between several bleomycin compounds with a stemloop DNA have been investigated by using NMR spectroscopy. The study provides important advances in our understanding for how the subtle changes in the bleomycin C-terminal tails might lead to larger changes in the efficacy and binding behaviour towards DNA targets containing the GT dinucleotide.

Although the experiments have been conducted properly and are well described, I have several concerns regarding the presentation of the results and aspects of the data interpretation.

A major concern is the overall presentation of the results within the main text. At present many of the analyses require switching between a figure in the main text, at least one table, and also data in the supplementary materials. In all cases, the addition of more annotation would greatly simplify this process. For example, the annotation style in Figure 8 should be applied to the other spectra, and where possible include actual nucleotide numbers, and ideally the atom as well. For Figure 2, the imino peaks in the unbound (blue) spectrum for OL2 should be annotated with nucleotide name/number. Since there is no other reference to peak number in the main text or tables (apart from in the figure panels), I think the peak number system should only be used within the supplementary materials, or used more explicitly in the main text. For busier spectra that have multiple NOE to a single atom, perhaps a line can be added to the spectra to show that these NOE belong to a single (of more) atom(s), and then the line can be annotated with the atom name/base number.

Related to Figures, I would suggest to remove Figure 3 and instead place the corresponding annotated spectral region of the unbound OL2 as panel (a) within Figure 4 and Figure 5. The updated Figure 4 would then have four panels, the first panel (a) of the unbound OL2 spectrum, only the H2O spectrum is required, with annotation of some of the major peaks, and panels b,c,d (that is, the old panels a,b,c) with annotation of the new peaks. Similary, an updated Figure 5 would have two extra panels at the top which would be (a) the unbound OL2 (H2O only) corresponding to the old Figure 3b, and (b) an annotated spectrum of unbound OL2 (H2O only) that relates to the second selected region of the spectra.

Figure 8 would also be improved by having an initial spectrum of the unbound OL2 inserted as panel (a). There would therefore be four spectra in this figure. The unbound OL2 spectrum would  have annotation of the (black) peaks, and then only annotate the bound (red) peaks in the remaining three spectra.

The same idea would continue with Figure 10, with a new panel (a) for the unbound OL2 (but this time the D2O spectrum)

Similarly, it would help if the image in Figure 6 was instead incorporated as a panel in Figure 7. The new Figure 7 would then have four panels for easier comparison. It would also help to move the unbound NOE network from Figure 9 as panel (a) in Figure 11.

Finally, Figure 2 could be improved by the addition of a simple schematic of the OL2 DNA.

In terms of data interpretation, it appears that loss of NOE peaks is often observed upon addition of the bleomycin derivatives. At some points this observation as interpreted simply as an effect on the DNA structure, and this would be the safest interpretation (such as in line 122,123). However, the loss of NOE peaks is most often explained as a change in conformation throughout the majority of the text and also in the title of the manuscript. It is equally possible that certain DNA peaks are instead more broadened upon binding bleomycin, and in this case a change in stability or dynamics would be another interpretation. The authors should comment on this possibility, and confirm that a change in conformation is the main consequence of bleomycin binding, or modify the text to indicate that the change can be in the conformation, dynamics or stability. This broadening is, for example, significant in the 1d spectra of Figure 2.

Also related to data interpretation, approximate affinities of the bleomycin for the OL2 DNA is not indicated in the introduction. This would be important to discriminate changes in binding behaviour due to the extent bound (affinity) versus the specific details of binding by the bleomycin C-terminal tails. The concentrations of the samples should also be provided in order to compare the sample concentrations with the approximate affinity constants in order to judge the extent of binding.

It should also be commented upon why no NOEs are observed between the bleomycin and the DNA throughout the manuscript, or if these NOEs might in some case relate to the new peaks observed in the bleomycin:OL2 complexes.

Minor comments:

- the 'prime' is missing in the title after the 5

- the use of the word 'triad' is sometimes confusing - more commonly this refers to three nucleotide bases interacting together. In this case, the word 'complex' might be used for clarity to refer to the two bases plus ligand

Author Response

Reviewer 4

Thank you very much for your comments.

I will cite this Reviewer’s comments textually in italics and bold, and follow with an answer to each of them.

Comment 1: “A major concern is the overall presentation of the results within the main text. At present many of the analyses require switching between a figure in the main text, at least one table, and also data in the supplementary materials. In all cases, the addition of more annotation would greatly simplify this process. For example, the annotation style in Figure 8 should be applied to the other spectra, and where possible include actual nucleotide numbers, and ideally the atom as well.

Answer 1: We will try to satisfy this request wherever possible. Additional annotation and figure consolidation could render the modified figures very difficult to see in terms of labels and NMR signals.

Comment 2: “For Figure 2, the imino peaks in the unbound (blue) spectrum for OL2 should be annotated with nucleotide name/number. Since there is no other reference to peak number in the main text or tables (apart from in the figure panels), I think the peak number system should only be used within the supplementary materials, or used more explicitly in the main text.

Answer 2: Assignments for these signals were added in Figure 2

Comment 3: “For busier spectra that have multiple NOE to a single atom, perhaps a line can be added to the spectra to show that these NOE belong to a single (of more) atom(s), and then the line can be annotated with the atom name/base number.

Answer 3: Doing what Reviewer 4 suggests would make the figure very difficult to read, since, although close, many of the NOEs that seem to be to the same atom are not. This means that multiple lines and annotations would have to be added, rendering the corresponding figure very busy.

Comment 4: “Related to Figures, I would suggest to remove Figure 3 and instead place the corresponding annotated spectral region of the unbound OL2 as panel (a) within Figure 4 and Figure 5. The updated Figure 4 would then have four panels, the first panel (a) of the unbound OL2 spectrum, only the H2O spectrum is required, with annotation of some of the major peaks, and panels b,c,d (that is, the old panels a,b,c) with annotation of the new peaks.

Answer 4: Unfortunately, adding another spectrum to Figures 4 and 5 would render the spectra and corresponding labels very small to read. Additionally, overlapping spectra (bound and free, black and red), already includes the spectra of free OL2 in each figure.

Comment 5: “Similarly, an updated Figure 5 would have two extra panels at the top which would be (a) the unbound OL2 (H2O only) corresponding to the old Figure 3b, and (b) an annotated spectrum of unbound OL2 (H2O only) that relates to the second selected region of the spectra.

Answer 5: Same answer as that to comment 4.

Comment 6: “Figure 8 would also be improved by having an initial spectrum of the unbound OL2 inserted as panel (a). There would therefore be four spectra in this figure. The unbound OL2 spectrum would have annotation of the (black) peaks, and then only annotate the bound (red) peaks in the remaining three spectra.

Answer 6: Same answer as that to comment 4.

Comment 7: “The same idea would continue with Figure 10, with a new panel (a) for the unbound OL2 (but this time the D2O spectrum)

Answer 7: Same answer as that to comment 4.

Comment 8: “Similarly, it would help if the image in Figure 6 was instead incorporated as a panel in Figure 7. The new Figure 7 would then have four panels for easier comparison.

Answer 8: This was done because the lines and labels can still be seen in the consolidated figure.

Comment 9: “It would also help to move the unbound NOE network from Figure 9 as panel (a) in Figure 11.

Answer 9: This cannot be done without making lines and labels very small and difficult to see.

Comment 10: “Finally, Figure 2 could be improved by the addition of a simple schematic of the OL2 DNA.

Answer 10: Done

Comment 11: “In terms of data interpretation, it appears that loss of NOE peaks is often observed upon addition of the bleomycin derivatives. At some points this observation as interpreted simply as an effect on the DNA structure, and this would be the safest interpretation (such as in line 122,123). However, the loss of NOE peaks is most often explained as a change in conformation throughout the majority of the text and also in the title of the manuscript. It is equally possible that certain DNA peaks are instead more broadened upon binding bleomycin, and in this case a change in stability or dynamics would be another interpretation. The authors should comment on this possibility, and confirm that a change in conformation is the main consequence of bleomycin binding, or modify the text to indicate that the change can be in the conformation, dynamics or stability. This broadening is, for example, significant in the 1d spectra of Figure 2.

Answer 11: Exchange effects were not explored for the binding of Zn(II)BLM-A5 to OL2 in the present manuscript. However, in our work with Zn(II)PEP and Zn(II)BLM-A2 bound to the same oligonucleotide (ref 21) ROESY spectra were collected to explore this possibility. The ROESY spectra collected for both triads exhibited no cross peaks due to exchange. Since all BLMs have very similar chemical structure and molecular weight, we assume that ROESY spectra collected for the A5 triad would render the same results. In the absence of exchange ROE signals, we interpret the missing NOE signals as a sign of changes in the conformation of OL2 upon MBLM complexation. A paragraph stating these results has been added in the reviewed manuscript as part of the discussion (216-223).

Comment 12: “Also related to data interpretation, approximate affinities of the bleomycin for the OL2 DNA is not indicated in the introduction. This would be important to discriminate changes in binding behaviour due to the extent bound (affinity) versus the specific details of binding by the bleomycin C-terminal tails. The concentrations of the samples should also be provided in order to compare the sample concentrations with the approximate affinity constants in order to judge the extent of binding.

Answer 12: The affinity of Zn(II)BLM-A5 for OL2 was not determined. However, the affinities of Zn(II)BLM-A2 and Zn(II)PEP for OL2 were determined in ref 21. They are: 6.7x10-6 M for the PEP triad and 3.1x10-6 M for the A2 triad.

Comment 13: “It should also be commented upon why no NOEs are observed between the bleomycin and the DNA throughout the manuscript, or if these NOEs might in some case relate to the new peaks observed in the bleomycin:OL2 complexes.

Answer 13: Intermolecular interactions in the form of NOEs were not detected either at 25 or 5 ºC. Intermolecular NOEs were detected by Manderville et al. for Zn(II)BLM bound to a DNA fragment containing a 5’-GC-3’ binding site only at 35 ºC but not at lower temperatures, and the detected intermolecular NOEs were very week signals. All of our spectra were collected at 25 or 5 ºC, which made us able to assign NH and NH2 signals. These signals cannot be assign at higher temperatures due to fast exchange with H2O (lines 224-229)

Comment 14: - “the 'prime' is missing in the title after the 5

Answer 14: Fixed. Thank you

Comment 15: “- the use of the word 'triad' is sometimes confusing - more commonly this refers to three nucleotide bases interacting together. In this case, the word 'complex' might be used for clarity to refer to the two bases plus ligand

Answer 15: Triad is defined in the manuscript (line 61). This term has been used in the series of four papers in the subject of Z(II)BLM bound to hairpins, and we would like to conserve it to keep consistency.

Round 2

Reviewer 1 Report

 The paper in the present form is suitable for publication.

Reviewer 2 Report

In my opinion, the amount of new experimental data included in the manuscript is insufficient for this article to appear in Magnetochemistry. According to the authors, most of the previously published data were needed  “to use that data to compare to what happens when Zn(II)BLM-A5 binds the same hairpin”. However, the discussion of these results does not bring any structurally significant conclusion.

I maintain my negative opinion and do not recommend this article for publication.